# Factors Affecting the Leaching of Chloropropanols from Pulp Used in the Manufacture of Paper Food Packaging

**DOI:** 10.3390/foods11152284

**Published:** 2022-07-30

**Authors:** Jinwei Zhao, Xin Wang, Jiao Li, Shuangquan Yao, Shuangfei Wang, Chen Liang, Chengrong Qin

**Affiliations:** 1College of Light Industry and Food Engineering, Guangxi University, Nanning 530004, China; 1916391038@st.gxu.edu.cn (J.Z.); 2016301031@st.gxu.edu.cn (X.W.); 2116391017@st.gxu.edu.cn (J.L.); yaoshuangquan@gxu.edu.cn (S.Y.); wangsf@gxu.edu.cn (S.W.); qin_chengrong@163.com (C.Q.); 2Guangxi Key Laboratory of Clean Pulp and Paper and Pollution Control, Guangxi University, Nanning 530004, China

**Keywords:** response surface analysis, chloropropanol, extraction, formaldehyde solution, paper material

## Abstract

Paper packaging materials are widely used in food packaging. However, it is difficult to extract trace chloropropanol from food packaging paper, so there is a lack of research on the leaching of chloropropanol from paper materials. Therefore, it is of positive significance to explore the leaching rule of chloropropanol in paper pulp for the safety of paper packaging materials, to reduce the risk of food packaging to food safety and to provide a theoretical basis for the formulation of safety standards for paper packaging materials. In order to study the content of chloropropanol in paper packaging paper more accurately, a response surface methodology was used to study the relationship between the amount of chloropropanol leaching and pulp concentration, leaching temperature and leaching time, as well as the interaction of each factor. The results showed that time, temperature and pulp concentration were the main factors affecting the leaching amount of chloropropanol from paper packaging materials. There were significant (*p* > 0.05) interactions between the time and pulp concentration, as well as temperature and pulp concentration. The leaching efficiency of chloropropanol was higher at a lower pulp concentration, and the leaching amount of chloropropanol was higher at a lower temperature. The temperature more significantly affected the leaching of chloropropanol in a low-concentration system than in a high-concentration system. Relevant studies have shown that the selection of solvent has an important effect on the extraction rate of target compounds. Therefore, in this experiment, different polar organic solvents (methanol, ethanol, formaldehyde solution) were added into the soaking solution to change the leaching amount of chloropropanol. The results showed that adding a certain amount of formaldehyde solution can obviously increase the leaching amount of chloropropanol in pulp.

## 1. Introduction

Paper packaging has become a better choice for food packaging because of its recyclable, biodegradable, renewable and sustainable characteristics [1]. Paper packaging to replace part of the transformation of plastic products is steadily advancing: the industry has formed paper bags and paper tableware as the representative of the alternative products. The safety of food and food packaging materials has been a hot topic for a long time. Food packaging materials are often manufactured with chemical additives such as sorbitol, glycerin, polyethylene glycol (PEG) and polyamide epichlorohydrin (PAE) to improve their function and protection [2]. However, chemical additives from packaging materials often migrate into food and lead to food safety issues. The European Union strictly limits the addition of chemical additives in packaging materials and restricts the quantity of chloropropanols. It is stipulated that 3-MCPD cannot exceed 12 μg/L and 1,3-DCP cannot exceed 2 μg/L. Therefore, the detection and source of all kinds of harmful substances is one of the important research contents in the field of food safety [3].

Chloropropanol is a highly biotoxic substance. Chlorpropanol pollutants such as 3-chloro-1, 2-propanediol (3-MCPD) and 1, 3-dichloro-propanol (1,3-DCP) have been detected in a variety of food and food packaging materials [4]. Chloropropanol is an internationally recognized food contaminant. Modern toxicological studies show that chloropropanol substances have a variety of toxicities to organisms. It will attack various organs of the organism, cause pathological changes in various tissues in the organism and affect the normal function of the organism. Available toxicity data suggest that both 3-MCPD and 1,3-DCP are carcinogens. They are classified as possible human carcinogens (Group 2B) [5].

With the introduction of various “plastic restriction” or even “plastic prohibition” policies, paper food packaging materials are increasingly appearing in people’s life. The quality detection and source research of harmful substances in paper products has become particularly important.

At present, most of the research on chloropropanol is focused on solving the problem of chloropropanol pollution in food, but also found chloropropanol substances in paper materials. Existing studies indicate that there are two main sources of organochlorides in paper materials. One is in the paper food packaging materials in the production process of various additives and the introduction of organic chloride. Polyamide epichlorohydrin (PAE) is often added to paper food packaging materials to improve the wet strength of paper materials [6]. This polymer is usually made from epichlorohydrin, of which, 3-MCPD and 1,3-DCP are by-products. The other is the existence of chlorinated organic compounds in the paper production system, which enter the material through adsorption and other ways [7]. Recent studies have found that chloropropanol can also be produced from paper materials during chlorine bleaching.

In paper food packaging materials, white cardboard plays an important role. White cardboard is a paper product made from bleached chemical pulp and high-yield pulp with a high stiffness and surface strength; it is widely used in food packaging materials. High-yield pulp is used in its production, such as sulfonated chemical mechanical pulp and alkaline hydrogen peroxide mechanical pulp. The former is rich in sulfonic acid groups [8], and the latter has more carboxyl groups [9]. These two groups are highly adsorbent; therefore, paper packaging materials are very likely to adsorb chemical additives and chlorinated organic compounds in bleaching wastewater in the production process, thus causing food safety risks.

Presently, chloropropanol is widely researched in connection with food and food additives [10]; however, methods for the extraction of chloropropanol from paper materials are not completely developed. Moreover, studies on the leaching of chloropropanol from pulp are considerably limited. The American Association of Official Analytical Chemists presents the most widely recognized method to determine chloropropanol in food and food additives. Chloropropanol is extracted from samples using cold water immersion [11]. In 2002, the German Federal Office for Consumer Protection and Food Safety proposed a standard method for extracting chloropropanol from paper packaging materials. Recently, numerous researchers have optimized the conventional experimental conditions deemed suitable for the compliance testing of chloropropanol migration. The derivatization was optimized to reduce the cost and time of analysis [4]. Zhong et al. explored the effect of water extraction conditions on the release of chloropropanol in food-contact paper and confirmed the effectiveness of cold water extraction. At high temperatures (80 °C or higher), the extraction yield of chloropropanols decreases, owing to volatilization losses [12]. Cold water extraction is currently recognized as the most effective method for extracting chloropropanol from solids. However, Djeridane used a 70% ethanol solution [13]. Diem Do used an aqueous methanol solution [14], and Xiao et al. used an aqueous formalin solution [15] for extracting active ingredients in solids, and obtained results that provide a reference for the efficient extraction of chloropropanols from paper materials.

In this study, the response surface analysis method was used to determine the relationship between the amount of chloropropanol leached from pulp and its soaking time, pulp concentration, soaking temperature and the interaction between the aforementioned three variables. Additionally, the effect of the addition of organic solvents such as methanol, formaldehyde and ethanol on the amount of chloropropanol leached from pulp was investigated. It is of positive significance to explore the leaching rule of chloropropanol in pulp, to explore the safety of paper packaging materials and to reduce the impact of food packaging on food safety risks.

## 2. Experimental Section

### 2.1. Experimental Instruments and Reagents

#### 2.1.1. Materials

Bleached softwood pulp boards were sourced from a paper mill in Guangxi, China.

#### 2.1.2. Experimental Equipment

GC–MS 5977A MSD (Agilent Technologies, Co., Ltd., Santa Clara, CA, USA), Dionex Autotrace 280 SHJ-6CS (Thermo, Waltham, MA, USA)

#### 2.1.3. Experimental Reagents

N-Heptafluorobutyrylimidazole (Sigma Reagent Company, Shanghai, China); n-hexane, ethyl acetate, methanol (chromatographically pure; Sigma Reagent Company, Shanghai, China); chloropropanol (3-Chloro-1-propanol, 98%; Sigma reagent company, Shanghai, China); hydrophilic-lipophilic-balanced (HLB) extraction cartridge (60 mg, 3 cc, Waters company, Milford, MA, USA); anhydrous calcium sulfate (99.99%; Sigma Reagent company, Shanghai, China) were used.

Standard solution: n-hexane was used as solvent to prepare 3-MCPD standard solutions of 0.002, 0.004, 0.006, 0.008 and 0.01 μg/mL, which were collected in screw-top glass bottles to be derivatized.

### 2.2. Experimental Steps

#### 2.2.1. Response Surface Design

Based on previous preliminary experiments and conclusions drawn by various researchers, it can be concluded that the method of extracting chloropropanol from paper materials, soaking temperature, soaking time and pulp concentration considerably affect the dissolution of chloropropanol in pulp. However, the influence of a single factor and the interactive influence between several factors remain unclear. Therefore, the soaking time, soaking temperature and pulp concentration were used as three-level factors while designing the response surface. In the method proposed by the German Federal Office for Consumer Protection and Food Safety for extracting chloropropanol from paper materials, the soaking time is supposed to be 24 h [16]). However, the dissolution of chloropropanol was found to continuously increase within 48 h. Therefore, to obtain the maximum dissolution amount of chloropropanol in pulp, the maximum soaking time was set to 72 h. According to previous research, the dissolution amount of chloropropanol for a soaking time of 2 h can meet the detection requirements; thus, the shortest soaking time was considered to be 2 h. Accordingly, the soaking time was set in the range of 2–72 h. Recently, Zhong et al. investigated the effect of temperature in the range of 10–60 °C on the leaching of chloropropanols [12]. As the sample must be shaken at a constant temperature for 2–72 h, laboratory testing is typically performed at room temperature; hence, the minimum temperature is 20 °C. To determine the effect of temperature on the leaching of chloropropanol from pulp, the selected temperature level was 20–60 °C. For extracting chloropropanol from pulp, 10 g of the pulp was generally added to 250 mL of the extractant. The pulp concentration was 0.04 g/mL. When the pulp concentration is greater than 0.12 g/mL, the amount of free water in the system is minimal. The extent of shaking is affected; therefore, the pulp concentration was approximately selected to be 0.02–0.12 g/mL.

The three-factor three-level response surface analysis was used to examine the interaction between the aforementioned three factors affecting the amount of chloropropanol leached from pulp. The designed response surface experiment is presented in Table 1.

#### 2.2.2. Immersion Extraction Treatment

The pulp boards sample was cut into 1 cm^2^ pieces with a pair of scissors. Quantitative samples were weighed with an accuracy of 0.001 g. The entire cardboard block was transferred to a 500 mL Erlenmeyer flask. A graduated cylinder was used to measure 250 mL of the soaking solvent, which was poured into a conical flask, and the mouth of the bottle was covered with a plastic wrap. The test sample was placed in a shaker. The temperature was set, and the sample was stirred evenly at 200 rpm to completely disperse the cardboard. The dispersed slurry was filtered with a 200-mesh polyester mesh. The filtrate was subsequently filtered with a 0.45 filter membrane to eliminate solids such as fine fibers, and the collected liquid samples were sealed and stored.

#### 2.2.3. Solid-Phase Extraction

The samples were extracted using HLB solid-phase extraction cartridges to separate chloropropanols. Subsequently, 5 mL each of ethyl acetate, methanol and ultrapure water was pipetted to activate the diatomite solid-phase extraction (SPE) extraction cartridge. Thereafter, 100 mL of the sample was measured to elute the extraction cartridge; the eluent was then discarded. The sample was subjected to blow drying with nitrogen for 20 min, and it was then eluted with 3 mL of n-hexane. The eluate was collected, concentrated to 1 mL by nitrogen blowing and transferred into a 10 mL screw-top glass bottle. Anhydrous calcium sulfate (1 g) was added, vortexed, mixed for 10 s and dehydrated [17].

#### 2.2.4. Sample Derivatization

Heptafluorobutyrylimidazole (50 μL) was added to the dehydrated sample, which was then vortexed for 10 s. The sample was incubated in a water bath at 70 °C for 20 min and then cooled to room temperature (25 °C). Subsequently, 5 mL of a saturated NaCl solution was added to the sample, which was again vortexed for 10 s and left to stand for stratification. The upper n-hexane organic phase was collected in a 10 mL screw-top glass bottle; anhydrous calcium sulfate (1 g) was added to the sample, vortexed for 10 s and extracted with a 1 mL medical syringe. Organic phase was filtered with a 0.22 μm organic filter membrane and stored in a 2 mL injection vial for gas chromatography–mass spectrometry (GC–MS) analysis [18,19].

#### 2.2.5. Quantitative Analysis by GC–MS

Conditions for GC–MS: column selection: an HP-5MS capillary column (30 m × 0.25 mm × 0.25 μm) with an inlet temperature of 220 °C was selected. The heating program was conducted as follows: the initial temperature was set to 40 °C; this was increased to 90 °C at 2 °C/min and maintained for 2 min. Subsequently, it was increased to 280 °C at 20 °C/min, and the apparatus was operated for 5 min at 280 °C. The carrier gas was helium (99.99% pure); the flow rate of helium was 1.0 mL/min for a splitless injection mode. The ion source specifications were as follows: ion source: electron bombardment (EI); ion source 230; Aux-2 temperature 280 °C; ionization energy; 70 eV; solvent delay: 3 min; selected ion monitoring mode, acquisition range: 35–175 *m*/*z* [19,20]. The characteristic peaks of the qualifier ions of chloropropanol were 41, 69, 77 and 169. The retention time was set to 7.5 min. The regression equation for the standard curve of chloropropanol is
$$y=2061.1x-0.5958;\;R^{2}=0.9998.$$

This equation indicates that the mass concentration of a chloropropanol standard solution demonstrates a good linear relationship with its peak area in the range of 0.002–0.008 μg·g^−1^, which satisfies the requirements for analysis. The formula for calculating the concentration of chloropropanol is obtained from the standard curve: $$C=\frac{A-B}{D}$$

*C*—Chloropropanol concentration;

*A*—Peak area;

*B*—3.16 × 10^6^;

*D*—1.123 × 10^10^.

Chloropropanol dissolution mass formula: m=C×M×V

*C*—Chloropropanol concentration;

*M*—Molecular weight of chloropropanol;

*V*—Volume of the extracted solution.

## 3. Results and Discussion

### 3.1. Response Surface Analysis and Interaction between Factors

The Design-Expert v.10 (Stat-Ease, Minneapolis, MN, USA) statistical software was used to analyze the regression model. The significance level and variance analysis results between factors are presented in Table 2. Both the F and *p* values represent the results of the significance test. The probability of Prob > F in the fitted equation is less than 0.0001, indicating that the model exhibits a significant effect. The lack of a fit item is 0.2352 > 0.05, indicating that its corresponding test is not obvious, and that the difference is not significant. This implies that the residuals result from random errors, and that the reliability of the equation is high. In this study, R^2^ = 0.9782; the fitting degree of the equation is higher than 90%, and the simulated regression equation can be used for prediction. The coefficient of variation (C.V.%) indicates the degree of dispersion of data. Low C.V. values indicate low average levels of the variable and a better fit than high C.V. values. The C.V. simulated in this study was 3.92%, indicating that the experimental simulation data are reliable.

The soaking time, pulp concentration and soaking temperature primarily affect the amount of chloropropanol leached from pulp. The influence of each factor on the leaching amount of chloropropanol is, however, nonlinear. Therefore, the Design-Expert software was used to perform multivariate fitting on the obtained results to evaluate the influence of the three factors on the leaching rate of chloropropanol. Accordingly, the binary multinomial regression equation of the leaching amount of chloropropanol (*Y*) in terms of the soaking time (*A*), soaking temperature (*B*) and pulp concentration (*C*) is as follows:$$\begin{array}{ll} Y & =+0.49533870497563+0.02730512773726\times A-4.36724675237881E-003\times B\\ & +1.38103283267637\times C-7.46023903571440E-005\times A\times B+8.54272926150121E\\ & -003\times A\times C+3.56204052966102E-003\times C\times B-2.15281183122449E-004\times A^{2}\\ & -4.63308133125009E-005\times B^{2}-0.79784585413674\times C^{2} \end{array}$$

In the simulated equation, the magnitude of the coefficient reflects the degree of influence of the factor on the response value. The coefficient of the quadratic term in the quadratic multivariate equation model is negative, indicating that the equation has a maximum value and can be optimized for analysis. The response surface analysis reveals that “P > F” < 0.001; thus, the factor response is significant, and A, C, B, AC, A^2^ and C^2^ demonstrate the most significant leaching amounts of chloropropanol. According to the F value of each influencing factor, the degree of significant influence of each factor on chloropropanol can be determined. The order of magnitude is as follows: C > A > B > C^2^ > A^2^ > AC; that is, pulp concentration > soaking time > soaking time squared > pulp concentration squared > soaking time squared > interaction between time and pulp concentration. When “*p* < 0.05”, the effect is not significant; consequently, we can conclude that the interaction between soaking time and soaking temperature is not significant [21]. Using Design-Expert v.10 to fit the quadratic multiple regression equation model, the three-dimensional response surface map and contour line between the leaching amount of chloropropanol and the two factors were obtained. The effect of the interaction between the two factors on the leaching of chloropropanol from pulp can be distinctly observed based on the changes in the response surface (Figure 1).

#### 3.1.1. Effects of Soaking Time and Temperature on the Leaching of Chlorohydrin

Figure 1a,b indicate that, when the soaking time and soaking temperature interact with each other and when the temperature is constant, the leaching amount of chloropropanol increases with an increasing time and then stabilizes after it increases to a certain extent. When the soaking time is constant, the leaching amount of chloropropanol decreases with an increase in the soaking temperature, and the interaction between the soaking time and temperature does not significantly affect the leaching of chloropropanol.

#### 3.1.2. Influence of Interaction between Pulp Concentration and Soaking Time on Chloropropanol Leaching

Figure 1c,d indicate that, when the pulp concentration and soaking time interact, the pulp concentration is constant, and the leaching amount of chloropropanol in the pulp increases with an increasing time. However, it remains unchanged after reaching a certain concentration. This is likely because the chloropropanol in the pulp and the chloropropanol in the solution attain a dynamic equilibrium during the leaching of chloropropanol; thus, the content of chloropropanol in the solution does not increase further. When the soaking time is fixed, for an increasing pulp concentration, the leaching amount of chloropropanol gradually increases and follows an upward trend; however, the leaching efficiency in the system decreases with the increasing pulp concentration. Figure 2 indicates that the amount of chloropropanol dissolved in a unit mass of pulp decreases with an increasing pulp concentration. Therefore, the dissolution efficiency of pulp per unit mass is higher for lower pulp concentrations, and the time required to attain equilibrium is lower; this is a suitable option for the efficient and rapid dissolution of chloropropanol. With a high-consistency pulp system, the total amount of leached chloropropanol is relatively high; however, the pulp concentration cannot be increased infinitely. Therefore, an appropriate pulp concentration is selected according to the requirement.

#### 3.1.3. Effects of Pulp Concentration and Soaking Temperature Interaction on Chloropropanol Leaching

Figure 1e,f indicate that, when the pulp concentration and soaking temperature interact and when the soaking temperature is constant, the leaching amount of chloropropanol increases with an increasing pulp concentration, as shown in Figure 3. When the pulp concentration is constant, the leaching amount of chloropropanol in the pulp decreases with an increasing temperature. For a constant pulp concentration, the leaching amount of chloropropanol in the pulp decreases with an increasing temperature. Figure 4 indicates that, with an increasing temperature, the leaching amount of chloropropanol in the pulp decreases. In the range of 20–60 °C, the highest leaching amount of chloropropanol is obtained at 20 °C, whereas the lowest amount is obtained at 60 °C, which is consistent with the results reported in previous studies. It is possible that part of 3-McPd volatilizes with the increase in temperature, leading to the decrease in extraction efficiency [12]. It is also probably due to the possibility that 3-MCPD and 1,3-DCP do not just undergo a simple hydrolysis, but possibly either react with a component of the paper sample by an unknown mechanism or are adsorbed or retained by the fibers [7]). The pulp concentration and the soaking temperature affecting the leaching of chloropropanol are also different (Figure 5). The leaching of chloropropanol is affected to a greater extent by the temperature at lower pulp concentrations than at higher pulp concentrations. Figure 5 is obtained by comparing the leaching amount of chloropropanol at 20 °C with that at 60 °C for the same pulp concentration. Moreover, it can be observed that, when the pulp concentration is lower than 0.06 g/mL, the effect of temperature is significant. However, the effect is weakened with an increasing pulp concentration. At a pulp concentration of 0.06 g/mL, the temperature does not substantially affect the leaching of chlorohydrin. When the pulp concentration is further increased, the effect of temperature in the range of 20–60 °C on the leaching amount of chlorohydrin is not apparent; this finding is consistent with that of previous studies. Therefore, cold water extraction is considered as the most effective method to extract chloropropanol [7], and the effect of temperature on the leaching of chloropropanol is more significant under the condition of lower pulp concentrations.

### 3.2. Effects of Different Solutions on the Leaching of Chloropropanol

In this study, the effects of adding methanol, ethanol and formaldehyde to the aqueous solution were investigated in terms of the amount of chloropropanol leached from the pulp. The experimental results are displayed in Figure 6. The amount of chloropropanol leached from the aqueous methanol and ethanol solutions with different concentrations remains virtually unchanged, and the concentration of chloropropanol remains constant at 0.36 μg/g, which is almost similar to the amount of chloropropanol leached from the aqueous solution (0.37 μg/g). However, the concentrations of chloropropanol leached from formaldehyde solutions of different concentrations are considerably different. When the concentration of formaldehyde is less than 15%, the leaching amount of chloropropanol increases significantly with an increasing formaldehyde concentration. The concentration of chloropropanol is significantly higher than that of chloropropanol leached from the aqueous solution (up to 0.69 μg/g). Chloropropanol molecules are adsorbed onto the pulp by physical adsorption. The driving force for adsorption stems from intermolecular hydrogen bonds. Formaldehyde can react with the surface of cellulose and the hydroxyl groups in chloropropanol. Thus, the sites on cellulose combined with chloropropanol are passivated, thereby reducing the adsorbed amount of chloropropanol on the surface of cellulose and increasing the number of free chloropropanol molecules in the solution. Thus, the dissolution of chloropropanol is increased [15]. Therefore, formaldehyde significantly affects the amount of chloropropanol leached from the pulp.

## 4. Conclusions

Based on the response surface design experiment, it can be concluded that the soaking time, soaking temperature and pulp concentration significantly affect the leaching amount of chloropropanol in pulp. For a low pulp concentration, the leaching efficiency of chloropropanol from the pulp is high, and the time required to attain the leaching equilibrium is reduced. Moreover, the leaching amount of chloropropanol in pulp increases at low temperatures. The interaction between the soaking time and temperature is not significant; however, the interactions between the soaking time and pulp concentration, as well as those between the soaking temperature and pulp concentration, are significant. The effect of temperature on the leaching of chloropropanol in a low-pulp consistency system is more significant than that in a high-pulp consistency system. When the pulp concentration is 0.02–0.12 g/mL, the leaching amount of chloropropanol lies in the range of 0.002–0.006 μg/g, which satisfies the requirements for detection. The content of chloropropanol in the sample far exceeds the production standard for chloropropanol in the paper packaging of various countries, so the content of chloropropanol in paper packaging should be paid close attention to, and the food safety problems caused by it should also be paid attention to.

In the leaching experiment of chloropropanol from pulp, the addition of a certain concentration of aqueous formaldehyde solution to the pulp can effectively increase the amount of chloropropanol leached from the pulp. Formaldehyde solutions positively influence the efficient leaching of chloropropanols from pulp. It seems that cold water is not the most effective leaching solvent for paper packaging materials. The true content of chloropropanol in paper packaging materials is much higher than the current detection value, and the chloropropanol in paper packaging materials is not completely leached due to the limitation of the leaching solvent itself. Therefore, the development of an efficient solvent for the extraction of chloropropanol from paper packaging materials has become the focus of the following research.

## Figures and Tables

**Figure 1 foods-11-02284-f001:**
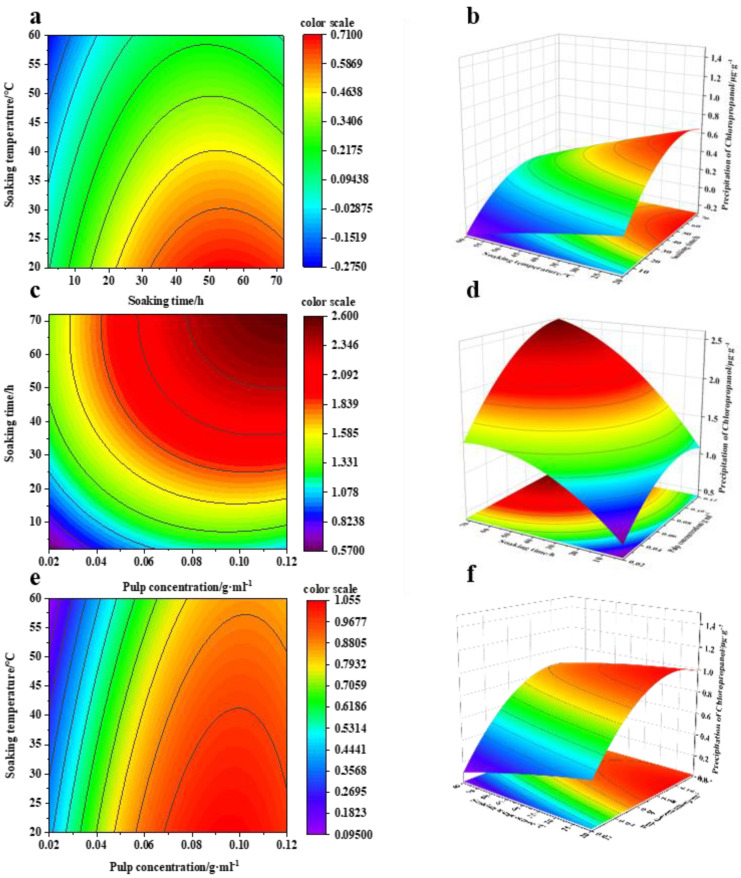
Three-dimensional response surface and contour lines between the dissolved amount of chloropropanol and the two factors. (**a**) The interactive effect of temperature and time, (**b**) 3D representation of (**a**), (**c**) Interaction of soaking time and pulp concentration, (**d**) 3D representation of (**c**), (**e**) Interaction of soaking temperature and pulp consistency, (**f**) 3D representation of (**e**).

**Figure 2 foods-11-02284-f002:**
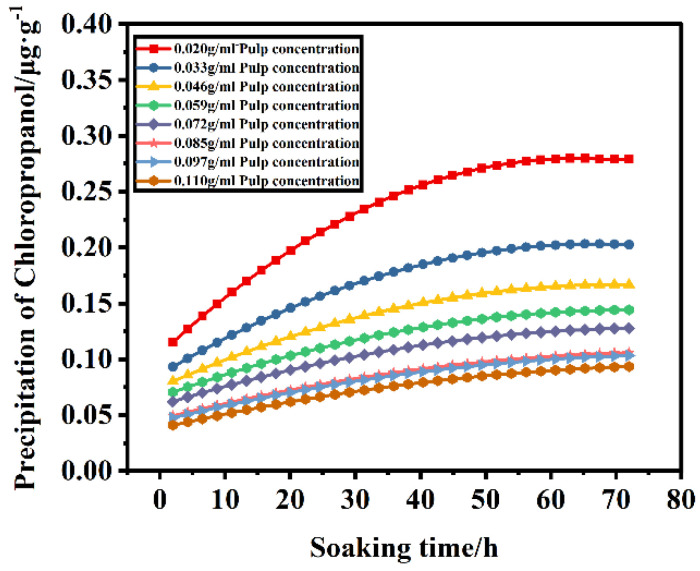
The amount of chloropropanol extracted from pulps with different concentrations varies with soaking time.

**Figure 3 foods-11-02284-f003:**
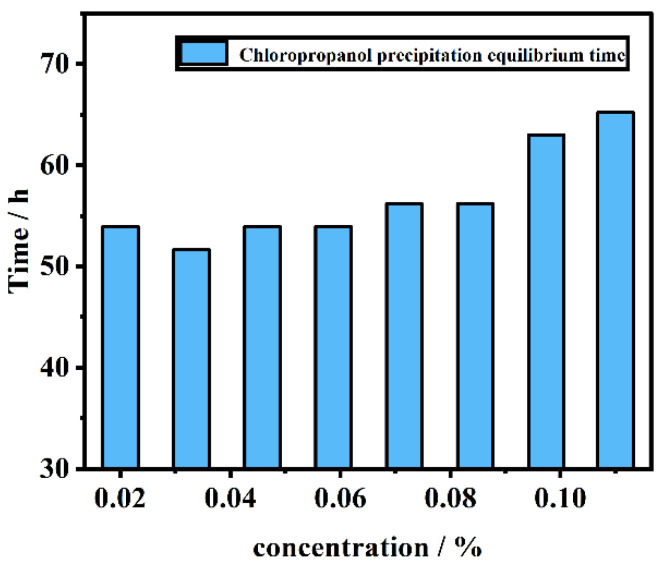
Chloropropanol leaching equilibration time.

**Figure 4 foods-11-02284-f004:**
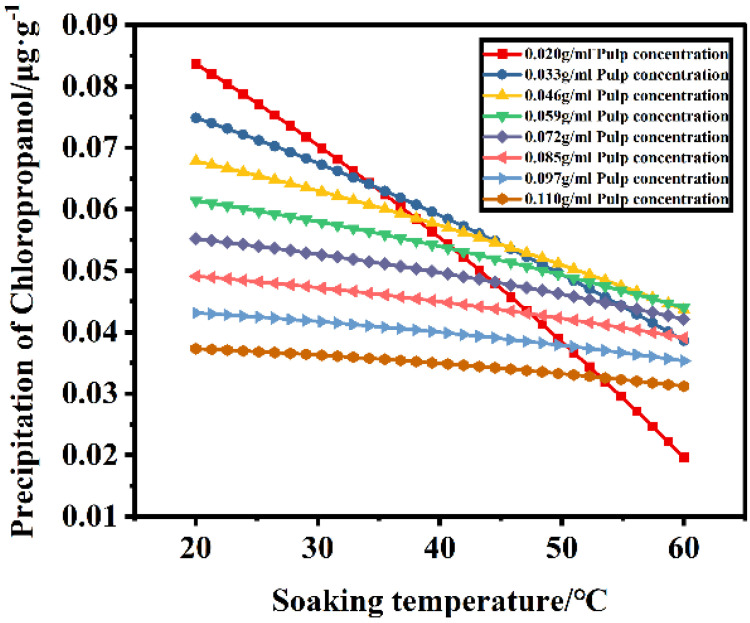
Influence of soaking temperature on the leaching amount of chloropropanol at different concentrations.

**Figure 5 foods-11-02284-f005:**
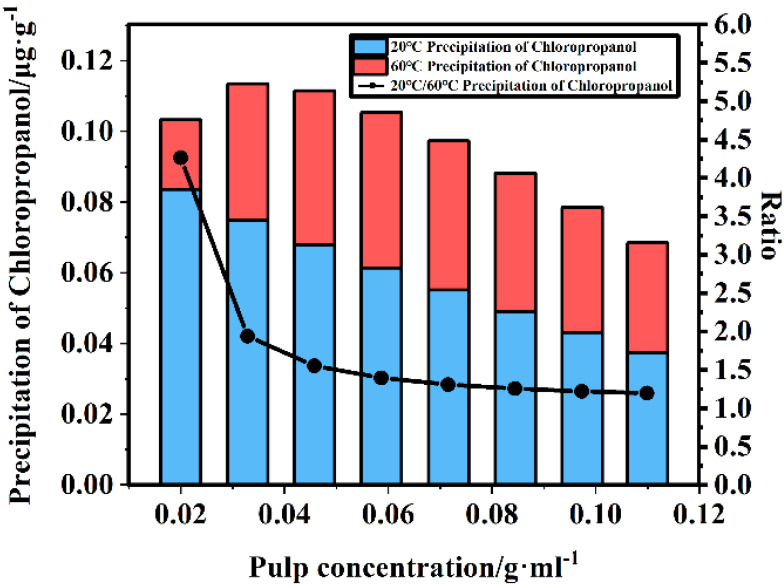
Leaching amounts of chloropropanol at 20 and 60 °C, respectively, and their ratio.

**Figure 6 foods-11-02284-f006:**
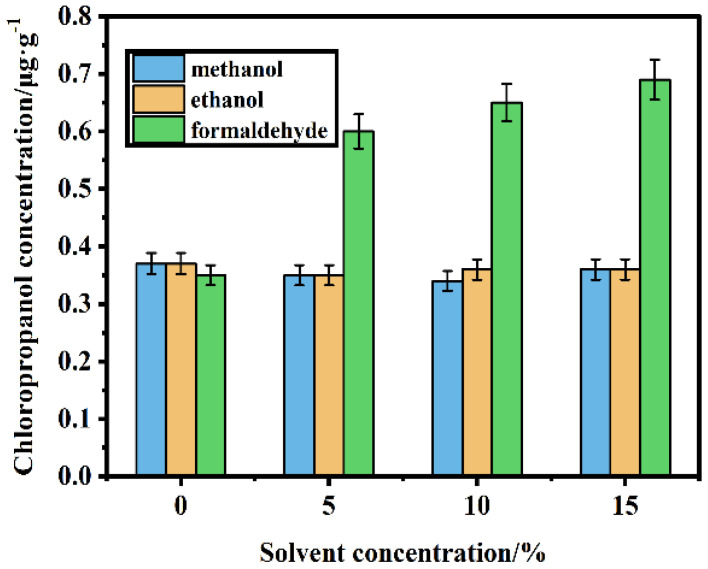
Leaching amounts of chloropropanol in methanol, ethanol and formaldehyde solutions.

**Table 1 foods-11-02284-t001:** Pulp concentration, temperature, time response surface three factors and level.

Factor	Level		
	−1	0	1
Pulp consistency	0.02	0.07	0.12
Temperature/°C	20	40	60
time/h	2	37	72

**Table 2 foods-11-02284-t002:** Results of three factors response surface analysis of pulp concentration, temperature and time.

	Sum of		Mean	F	*p*-Value	
Source	Squares	df	Square	Value	Prob > F	
Model	4.95	9	0.55	288.80	<0.0001	significant
*A—time*	1.81	1	1.81	952.56	<0.0001	
*B—temperature*	0.24	1	0.24	126.13	<0.0001	
*C—concentration*	2.09	1	2.09	1097.89	<0.0001	
*AB*	0.011	1	0.011	5.73	0.0479	
*AC*	0.12	1	0.12	65.40	<0.0001	
*BC*	7.067 × 10^−3^	1	7.067 × 10^−3^	3.71	0.0954	
*A* ^2^	0.29	1	0.29	153.86	<0.0001	
*B* ^2^	1.446 × 10^−3^	1	1.446 × 10^−3^	0.76	0.4123	
*C* ^2^	0.32	1	0.32	170.64	<0.0001	
Residual	0.013	7	1.903 × 10^−3^			
*Lack of Fit*	9.943 × 10^−3^	3	3.314 × 10^−3^	3.92	0.1099	*not significant*
*Pure Error*	3.380 × 10^−3^	4	8.449 × 10^−4^			
Cor Total	4.96	16				

## Data Availability

The data used to support the findings of this study can be made available by the corresponding author upon request.

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
