# Peer review of "Factors Affecting the Leaching of Chloropropanols from Pulp Used in the Manufacture of Paper Food Packaging"

_foods, 2022, doi:10.3390/foods11152284_

Round 1

Reviewer 1 Report

foods-1804515

Factors affecting the leaching of chloropropanols from pulp used in the manufacture of paper food packaging

Dear Authors,

The manuscript deals with the determination of factors affecting the leaching of chloropropanols from pulp used in the manufacture of paper food packaging. The manuscript has been well designed and written. It needs some necessary corrections. Comments are below;

-       Line 44: Give some samples for this sentence!

-       Please explain the formation ways of the chloropropanols!

-       Please give more information about the chloropropanols

-       Line 107: How was the soaking solution prepared?

-       Line 114: Give more information about the HLB cartridge

-       Lines 153 and 154: How were these values determined?

-       Method validation parameters should be added into text.

Author Response

 - 1.Line 44: Give some samples for this sentence!

I have added them in the article.

-2. Please explain the formation ways of the chloropropanols!

There are two main sources of organochlorides in paper materials. One is in the paper food packaging materials in the production process of various additives and the introduction of organic chloride. Polyamide epichlorohydrin (PAE) is often added to paper food packaging materials to improve the wet strength of paper materials. This polymer is usually made from epichlorohydrin, of which 3-MCPD and 1,3-DCP are by products. The other is the existence of chlorinated organic compounds in the paper production system, which enter the material through adsorption and other ways. Recent studies have found that chloropropanol can also be produced from paper materials during chlorine bleaching.

-3.Please give more information about the chloropropanols

I have added them in the article.

-4.Line 107: How was the soaking solution prepared?

In the experiment of chlorpropyl alcohol leaching of paper materials, distilled water is used as the soaking solvent. The standard solution with methanol, ethanol and formaldehyde was used in the experiment to explore the influence of organic solvents on the leaching amount of chloropropanol from paper materials.

-5.Line 114: Give more information about the HLB cartridge

It's described in the material. Hydrophilic-lipophilic-balanced (HLB) extraction cartridge (60 mg, 3 cc, Waters company, USA)

-6.Lines 153 and 154: How were these values determined?

Reference was made to the GC-MS detection method of chloropropanol used by

researchers before

-7.Method validation parameters should be added into text.

Analysis is made in 3.2 Response surface Analysis and interaction among factors

Reviewer 2 Report

This paper describes the method relatively well and compares few key factors. However, the background linking to paper making is described too shortly. The relationship, and impact of findings to food packaging is not discussed sufficiently.

Line 46-48.  Please provide the reference for this claim as this limit would be important information for other researchers as well.

Lines 53-55. The cited reference (Wong, Low and Khor 2020) does not seem to match the sentence. Please provide a reference that is more relevant to the claim. Also add a references to the recent studies stated at the end of line 55.

The introduction continues to state several claims about the chloropropanol food safety and analysis methods, but without the appropriate references.

Experimental section: Line 88: Please confirm the correct terminology – What is meant by paddle board?  Please describe in more detail if the sample was a pulp sheet or cardboard as in was is a sample of raw material from the pulp mill or was it initially already made into packaging material and then disintegrated. This should be absolutely clarified as it has major consequence on interpretation of the results. The terminology should be kept consistent.

The main variables of the study; soaking time, pulp concentration, soaking temperature have not been indicated anywhere in the experimental section. However the results and discussion part seems to start with some description including them. This part should be clarified and moved to the experimental section. Please add a citation to the  German Federal Office for 169 Consumer Protection and Food Safety method mentioned on line 169-170.

Results and discussion:

Table 2 needs an improvement – the table is unclear to the reader, the table caption needs to be improved so that it will be understood even when the table would be separated from the body text of the article. It would be good to link the naming of the variable parameters also to table 1 to keep consistency throughout the paper.

Line 226: Please check the formating here: ”…soaking time squared > interaction between time and pulp.” Is the symbol here correct as the abbreviated version above states AC for this part?

Figure 2 caption could be improved to give more information to the reader on what the reader should look for in the graphs.

Lines 241-243:” leaching amount of chloropropanol decreases with an increase in the soaking temperature” This result should be discussed thoroughly with references to literature. Most chemical reactions speed up or intensify as the reaction temperature is increasing. What is the reason for the decrease? There is comparison to previous work, but it remains unclear to the reader what is the reason behind this behavior. Therefore also Please also discuss more the meaning of interaction of soaking time and soaking temperature: what does it really mean in this context?

3.3. Effects of different solutions on the leaching of chloropropanol

What is the reason for selecting these solvents? From the food and packaging material perspective the use of ethanol can be justified as it is used as a model solvent (although mainly in 10 or 50% concentration) in migration testing. But why methanol and formaldehyde? What information that would be useful for food safety can be attained from this?

The paper starts with strong references to food safety, but the discussion and conclusions do not relate the results back to these. What is the significance of these results in respect to the food safety or food packaging? 

Author Response

1.This paper describes the method relatively well and compares few key factors. However, the background linking to paper making is described too shortly. The relationship, and impact of findings to food packaging is not discussed sufficiently.

I have added them in the article.

2.Line 46-48.  Please provide the reference for this claim as this limit would be important information for other researchers as well.

Corresponding values have been added in the article

3.Lines 53-55. The cited reference (Wong, Low and Khor 2020) does not seem to match the sentence. Please provide a reference that is more relevant to the claim. Also add a references to the recent studies stated at the end of line 55.

It has been modified in the article.

4.The introduction continues to state several claims about the chloropropanol food safety and analysis methods, but without the appropriate references.

References have been added to the paper.

5.Experimental section: Line 88: Please confirm the correct terminology – What is meant by paddle board?  Please describe in more detail if the sample was a pulp sheet or cardboard as in was is a sample of raw material from the pulp mill or was it initially already made into packaging material and then disintegrated. This should be absolutely clarified as it has major consequence on interpretation of the results. The terminology should be kept consistent.

It has been modified in the article.

6.The main variables of the study; soaking time, pulp concentration, soaking temperature have not been indicated anywhere in the experimental section. However the results and discussion part seems to start with some description including them. This part should be clarified and moved to the experimental section. Please add a citation to the  German Federal Office for 169 Consumer Protection and Food Safety method mentioned on line 169-170.

Has been added to the article.

7.Table 2 needs an improvement – the table is unclear to the reader, the table caption needs to be improved so that it will be understood even when the table would be separated from the body text of the article. It would be good to link the naming of the variable parameters also to table 1 to keep consistency throughout the paper.

It has been modified in the article.

7.Line 226: Please check the formating here: ”…soaking time squared > interaction between time and pulp.” Is the symbol here correct as the abbreviated version above states AC for this part?

The A in AC stands for pulp concentration,The C for soaking time, Soaking time squared > Interaction between time and pulp is the result of variable analysis in response surface analysis software.

8.Figure 2 caption could be improved to give more information to the reader on what the reader should look for in the graphs.

It has been modified in the article.

9.Lines 241-243:” leaching amount of chloropropanol decreases with an increase in the soaking temperature” This result should be discussed thoroughly with references to literature. Most chemical reactions speed up or intensify as the reaction temperature is increasing. What is the reason for the decrease? There is comparison to previous work, but it remains unclear to the reader what is the reason behind this behavior. Therefore also Please also discuss more the meaning of interaction of soaking time and soaking temperature: what does it really mean in this context?

 The paper has been modified to add the analysis of temperature increase and efficiency reduction.

3.3. Effects of different solutions on the leaching of chloropropanol

9.What is the reason for selecting these solvents? From the food and packaging material perspective the use of ethanol can be justified as it is used as a model solvent (although mainly in 10 or 50% concentration) in migration testing. But why methanol and formaldehyde? What information that would be useful for food safety can be attained from this?

The paper starts with strong references to food safety, but the discussion and conclusions do not relate the results back to these. What is the significance of these results in respect to the food safety or food packaging? 

Methanol, formaldehyde, ethanol and other organic solvents were selected for experiments, which were introduced in the introduction. Some researchers found that these solvents had achieved certain effects in the study of solid leaching active ingredients. Therefore, this paper explores the influence of these organic solvents on the leaching amount of chlorpropanol in paper materials. The experimental results show that formaldehyde can effectively increase the amount of chlorpropanol leaching from paper packaging materials, which indicates that water as a solvent can not completely leach chlorpropanol from paper packaging materials, which will cause the detection error of chlorpropanol in paper packaging materials.

Reviewer 3 Report

The manuscript titled “Factors Affecting the Leaching of Chloropropanols from Pulp 2 used in the Manufacture of Paper Food Packaging” by Zhao et al. evaluated the effect of various factors i.e., pulp concentration, soaking temperature, soaking time on leaching of chloropropanols. This manuscript tries to address the food safety problems caused by chloropropanols. While the overall work is interesting, it suffers several drawbacks that can probably prevent it from attracting a wide readership. Hence, the following queries/questions need to be addressed before the manuscript can be considered suitable for publication:

1.    The abstract must be thoroughly improved. The authors should mention the quantitative information on results obtained. The abstract must reflect the whole work plan and important results. The readers read the abstract to decide whether it is worth reading the article. Hence, it must be improved.

2.    The introduction is written poorly. The authors are unable to effectively discuss the details of the background study. As this manuscript deals with the problems associated with paper packaging, the authors provide unnecessary background related to plastic packaging. Further, the novelty of the current study is not mentioned clearly.

3.    The main problem with this article is in the statistical analysis. The authors must perform the statistical analysis using "Analysis of variance (ANOVA)”. The mean and standard deviation of the obtained data sets must be mentioned.

4.    The conclusion section is weak. Essential details about the obtained results are missing. It can be made stronger by adding some insights obtained from the results.

Author Response

  1. The abstract must be thoroughly improved. The authors should mention the quantitative information on results obtained. The abstract must reflect the whole work plan and important results. The readers read the abstract to decide whether it is worth reading the article. Hence, it must be improved.

It has been thoroughly improved in the manuscript.

  1. The introduction is written poorly. The authors are unable to effectively discuss the details of the background study. As this manuscript deals with the problems associated with paper packaging, the authors provide unnecessary background related to plastic packaging. Further, the novelty of the current study is not mentioned clearly.

It has been thoroughly improved in the manuscript.

  1. The main problem with this article is in the statistical analysis. The authors must perform the statistical analysis using "Analysis of variance (ANOVA)”. The mean and standard deviation of the obtained data sets must be mentioned.

 Response surface is to simulate the real limit state surface by fitting one response surface through a series of deterministic "tests".The basic idea is to assume that an analytic expression between the limit state function and the basic variables with some unknown parameters replaces the actual structural limit state function which cannot be expressed explicitly.Response surface analysis mainly analyzes variance, which is analyzed in detail in 3.2 Response surface analysis and interaction among factors.

  1. The conclusion section is weak. Essential details about the obtained results are missing. It can be made stronger by adding some insights obtained from the results.

It has been thoroughly improved in the manuscript.

Round 2

Reviewer 2 Report

Will require an edit to check for punctuation, spacing and minor grammar eg. lines 43, 46, 49

Statement on lines 69-70 does not indicate if the recent studies are the ones referenced in the paragraph or some other. Is this due to grammar or lacking the studies as references?

White card paper is not accurate terminology. The paper they are referencing refers to liquid packaging board, which is coated with plastic from the both sides. Please use (white) paperboard when referring to the finished product. (pulp when discussing fibers dispersed in liquid)

the point nr 5 from initial review has not been sufficiently addressed. Current line 112 has been corrected but the corresponding terminology changes to experimental section 2.2.1. are lacking.

point 6 in initial review feedback has not been taken into account. ”The main variables of the study; soaking time, pulp concentration, soaking temperature have not been indicated anywhere in the experimental section. However the results and discussion part seems to start with some description including them. This part should be clarified and moved to the experimental section.”

To state this more clearly: paragraph 3.1. Response surface design: at least the parts of lines 205-216 (incl. table 1) should be moved to experimental section (heading 2.x).

The discussion part here has mixed information – some of it could belong to the experimental part as well. The authors should reconsider that too.

Point 7 in initial review has been addressed hastily.

A key point in discussion and conclusions is still missing. Therefore it would be needed to critically consider the food safety aspect and the concept of migration of chemical components from food packaging materials to food. Why would it important to be able to extract all of the chloropropanol? Isn’t the point of migration studies to show what amount of chemical does migrate in the conditions that are relevant for the specific food (time, temperature and chemical composition of food in contact) considering the worst case scenario? As one is extracting chloropropanol now from the pulp itself in a solution this is already a very extreme case. The discussion about this is lacking in the paper. How do the solvents selected here relate to the model solvents that are used for migration studies? What new information that is important for a food scientist can be concluded from the results?  As Foods is the publishing journal, the food (interactions) aspect should be brought into more focus in the discussion and conclusions.

Author Response

I have answered your questions in turn in the following files。

Reviewer 3 Report

The manuscript titled “Factors Affecting the Leaching of Chloropropanols from Pulp 2 used in the Manufacture of Paper Food Packaging” has been improved in response to reviewers’ comments. However, a few queries need to be addressed.

1.    Please mention how many parallel samples are evaluated for determining the impact of variable factors on chloropropanols leaching. Mention error bars and significant differences in figures 2-6 (Wherever applicable).

Author Response

Three parallel experiments were carried out for each sample, and the data input into the response surface analysis software was the average value of the three experiments.

The data in Figure 2-5 are exported after analysis by the response surface software and have been processed, so there is no need to add error bars.
